# Military Family-Centred Resilience-Building Programming Across the Deployment Cycle: A Scoping Review

**DOI:** 10.3390/ijerph21101378

**Published:** 2024-10-18

**Authors:** Michèle L. Hébert, Joshua M. Tippe, Carley Aquin, Melody Maximos, Suzette Brémault-Phillips, Phillip R. Sevigny

**Affiliations:** 1Faculty of Rehabilitation Medicine, Department of Occupational Therapy, University of Alberta, Edmonton, AB T6G 2G4, Canada; maximosmelody@gmail.com; 2Heroes in Mind Advocacy and Research Consortium (HiMARC), Edmonton, AB T6G 2G4, Canada; tippe@ualberta.ca (J.M.T.); caquin@ualberta.ca (C.A.); 3Department of Educational Psychology, Faculty of Education, University of Alberta, Edmonton, AB T6G 2R3, Canada

**Keywords:** resilience program, resilience intervention, military family, parent, military service member, children and youth, mental health, wellbeing, scoping review

## Abstract

**Background**: There is international agreement that military families (MFs)—active service members, reservists, veterans, and their families—must be resilient to overcome military life adversities. Resilience is defined either as skillsets or as processes implicating multi-systems in a socio-ecological context. While research on resilience-building specific to children and families who face adversity is growing, there is a paucity of evidence on MF-centred resilience-building. **Objective**: This review describes the evidence on such resilience-building programming and determines if adversity is considered a barrier or facilitator to resilience-building. **Methods**: This scoping review yielded 4050 peer-reviewed articles from database inception until December 2023, found in 12 databases. Articles were deduplicated, leaving 1317 that were independently screened for eligibility by two reviewers. Disagreements were resolved through discussion with a third reviewer. **Findings**: Of these articles, 27 were included; 5 additional articles were also included from other sources. The vast majority of included studies (91%) were conducted in the United States. These 32 articles were organised into categories, including demographics, research methodologies used, resilience program descriptors, and outcomes. **Conclusions**: Our results reveal that programs on building MF resilience vary widely, often measuring non-resilience health and social outcomes. We provide preliminary insights for MF health and policy. Our review findings will be invaluable for further evidence-based programming that builds resilience in MFs.

## 1. Introduction

In this article, we present findings from a scoping review on programs that aim to build resilience in military families (MFs). Notably, MFs are defined differently in Australia, Canada, the United Kingdom and the US, respectively [1]. In this scoping review, we define MFs broadly as including any married or cohabiting couples, spouses or partners, elders, single parents or co-parents (whether cohabiting or living separately), and children from birth to young adulthood who are dependent on their parents or live at home. This definition is inclusive of active service members and reservists. We have chosen this broad child age range in light of neurodiversity considerations that may require an individual to depend on adult care from childhood through to adulthood. In Canada, approximately 163,000 MF members [2] form the backbone of the Canadian Armed Forces (CAF). The vast majority (80%) of Canadian military members have a civilian or stay-at-home-partner [3]. Almost half of military members (47%) and about a third of veterans (30%) have children younger than age 17 years, and almost half of military family members are children and teens (41%) [2]. Recently, the Canadian government affirmed MFs as making a positive contribution to the functioning of active-duty service members (SMs). Today, the CAF recognises the labour and loyalty of MF spouses and members, who are not active-duty SMs, yet still contribute to the military operational readiness of SMs to deploy at short notice [4]. This formal acknowledgement referred to MFs being “the strength behind the uniform” [4]. Other researchers highlight MF strength in deployment-related separation, where separations increased SMs’ appreciation for their spouse(s), as demonstrated by making intentional efforts for closeness [5].

Conversely, the health and family dynamics of MFs can be negatively impacted by military life. Manser [2] described MF-specific adversity by presenting six predominant and unique mental health issues in these families. Of these challenges, half are related to the military deployment cycle and the other half to family-related concerns. *Deployment cycle challenges* included (1) geographic relocations due to postings, (2) absences and separations from family due to operational requirements, and (3) operation-related illness, injury, or death. *Family-related challenges* included (4) personal wellbeing and health, (5) financial stability, and (6) relational intimacy with their partner. While these former three areas may seem relevant or similar to non-military family challenges, all six challenges emerge specifically due to military life unpredictability [2].

MFs also experience various mental health concerns [6,7,8,9,10]. Manser and colleagues [7] surveyed providers at 34 Military Family Resource Centres (MFRCs) in Canada, Europe, and the United States and identified the following three most common issues: (1) couple or family relationship difficulties, (2) child/youth mental health issues, and (3) difficulties with transitions/adjustment. Separation/divorce, family violence, lack of caregiver support, as well as a host of mood, anxiety, and substance use disorders, were also noted [7]. More than half the MFs surveyed (58%) reported that these mental health issues, while causing mild and reversible distress, resulted in functional impairments in MF members’ cognitive, social, and affective domains of life [7]. For example, an experience-sampling study with a sample of 254 veterans found that greater posttraumatic stress symptoms were associated with greater affective reactivity and lability in response to negative events [8]. Adolescents in MFs have also been found to report greater mental health symptoms than non-military youth [9], with the rate of military-related relocations potentially playing a contributing role in this trend [10]. Indeed, researchers agree that transitions and separations due to military deployment cycles are widely found to impact family dynamics and individual family member wellbeing [7,11,12].

Relocations and transitions back to civilian life present the most important challenges in civilian spouses and military SMs or veterans, respectively. While pre-deployment and operational readiness are central to active-duty SMs, relocations and transitions are widely experienced throughout the deployment cycle. For example, one quarter of Canadian military SMs must relocate for a new posting, rendering Canadian MFs three to four times more likely to relocate to a new province or territory in Canada than non-military families [13]. With each relocation, MFs must make difficult or significant life-changing decisions regarding spouse employment and children schooling, and they must seek new family doctors and community resources [2,14,15]. Relocations can require career sacrifices from spouses (51%) and, in more than half of MFs, relocations worsen their financial situation (52%) [2]. Indeed, relocations are a chief challenge for civilian spouses of military members [14]. Other military transitions, such as reintegration back to civilian family life, present additional challenges, particularly for the active-duty military SM or veteran who returns from military service. In some instances, military members transition into post-secondary student roles, and this transition type can be a difficult adjustment [15]. Along similar lines, Lee and colleagues [16] have noted a lack of readiness in both military members and their civilian spouses/partners for when veterans transition back to civilian life. However, veterans reported fewer perceived transitional challenges when their civilian spouse reported (a) an ability to manage stress, (b) access to social support, and (c) a sense of belonging [16].

Mental health and transitional difficulties extend to children and youth in MF, and these challenges are compounded when child neurodivergence is present. Despite being at increased risk for low wellbeing, hopelessness, and risk-taking behaviours [17], children and youth from military families are more likely to see family physicians for mental health services [18,19] but equally or less likely to use other publicly available mental health services [18]. Further, MFs who either have neurodiverse children or care for elderly parents experience compounding effects from the tumultuousness of military life [2]. In a US national report, few MFs with a neurodiverse child (19%) reported finding adequate childcare, and MFs with a neurodiverse family member reported that the process of seeking help for specialised services and continuity of care was their second most important need after mental health concerns [20]. Notably, in about one in ten MFs in Canada who have a neurodiverse child, the typical childcare work hours and the inconsistent childcare responsiveness to emergency and irregular demands from military life cause deleterious family burden [21]. Additionally, military parents and children/youth with neurodiversity may face compounded challenges due to deployment-related separations [22], combined with challenges obtaining disability-specific help [2]. Of note, voluntary separation was frequently reported in MFs with neurodiverse children living in the United States, among whom two thirds cited their “children’s education” as a central reason to choose separation over relocation [23].

While the majority of MFs describe themselves as responding well to military life challenges, many still experience struggles in adapting to these challenges [2,21]. From the perspective of respondents to a MFRC survey, the most significant gaps in military mental health services for MFs were in the domains of (1) peer support, (2) outreach and engagement services offered by the Canadian Armed Forces (CAF), and (3) mental health treatment services (i.e., psychological treatments for a mental health condition) [7]. The most significant challenges arose when any of the aforementioned stressors compound each other at any given time, also including single parenthood and dual-serving couples [21,22]. This context can create a significant amount of distress and fragmentation within MFs [2]. In sum, the unique difficulties faced in military life place a distinct and at times cumulative strain on military serving members, their partners, and their children. Military life stressors that at times are added to neurodiversity, ageing or other complex life factors, demand significant adaptability and responsiveness from military members and their families. Thus, the general consensus is that Canadian military serving members, veterans, and their families must be resilient so they can adapt to, bounce forward from, and/or thrive amid the challenges endemic to military life.

Moreover, the unprecedented COVID-19 pandemic intensified stressors that may have a compounding effect on MFs [24]. In addition to navigating the challenges inherent to military life, many MFs were faced with difficulties coping given pandemic-related changes to finances, work, routine, and childcare [25,26]. In a national survey conducted during the pandemic, in comparison to civilian families, 50% of MFs reported reduced income and/or civilian job loss, and another 50% reported a reduction in or loss of childcare [27]. Further, Urbieta and colleagues [28] found a compounding impact on MFs with neurodivergent children, where almost half (45%) of their parents reported difficulties maintaining special services support during the pandemic. Namely, a fair majority (65%) of military parents noted behavioural changes in their neurodivergent children due to forced COVID-related distancing, in turn causing social isolation [28]. Clinically significant anxiety and depression in MFs during the pandemic have been predicted by changes in work (due to childcare), household finances, and coping difficulties in MFs [25]. Furthermore, Keares and colleagues [29] found that use of medications to treat depression and identified mental health disorders increased in military children and adolescents during the COVID-19 pandemic. Altogether, these findings on COVID-related impact emphasise the importance for effective, hybrid or HyFlex MF-centred resilience-building programs.

Resilience has shifted from being understood as an attitude, skill or behaviour to being a complex and dynamic multi-systems process [30]. While global evidence on resilience-building programs is growing, little is currently known about the impact of such programs on MF resilience. Individual resilience skills and capacities are associated with many positive physical and mental health outcomes. When an individual adapts well to one stressor, they cultivate skills and capacities that can help them adapt to subsequent stressors [31,32,33]. More resilient individuals tend to enjoy more positive social, emotional, cognitive, and academic outcomes [34], and greater resilience also appears to reduce the impact of negative life events, buttress against negative life events, and improve a persons’ ability to adapt to future negative life events [35]. For example, Daly [34] reports that a *hardy attitude—*the quality of “being accustomed to dealing with fatigue or hardship” (p. 330)—can help individuals use the necessary internal (e.g., deliberation and careful planning) and external resources (e.g., social support) to overcome stress and adversity. For resilience in children, some propose that there are seven essential and correlated individual components that depict resilience—competence, confidence, connection, character, contribution, coping and control [36].

From a systems perspective, Masten and colleagues [30] define *resilience* “as the capacity of a dynamic system to adapt successfully through multisystem processes to challenges that threaten the function, survival, or development of the system” (p. 524). Family-level processes—beliefs, resourcefulness, and communication/problem-solving—are critical for helping families adapt to and overcome adversity [37,38]. Further, it is well-established that the family system—particularly parents—is central for developing children’s resilience, wellbeing, and social–emotional capacities and outcomes across the lifespan [39,40,41,42,43,44,45,46,47,48,49,50,51,52]. Similarly, systems perspectives emphasise how many factors and processes at multiple levels of analysis probabilistically interact to influence risk, resilience, and/or developmental outcomes over time [30,31,37,38,40,53,54,55,56,57,58,59,60]. Overall, multisystemic viewpoints of risk for developing psychopathology and resilience have become prominent in contemporary developmental psychology and resilience science [40,58,61,62]. Therefore, in this study, we define resilience as a system’s process of adaptation (e.g., individuals, groups, organizations) following adversity or risk exposure. Resilience is a dynamic process influenced by socio-ecological system interactions and available resources, which can function as protective factors and then lead to improved outcomes when coping with and overcoming adversity [60].

Governments are well positioned to contribute to MF resilience-building. MFs must navigate dual care systems when seeking help from both military- and civilian-specific programs. The resources at the disposal of military members and veterans, and the military hierarchies and chains of command, offer national military organisations opportunities to develop and implement evidence-based programs or interventions that promote resilience in MFs [63]. While MF supports exist, in one study, at least one-third of military spouses believed their national military organisation provided insufficient support to meet their needs [63,64]. Thus, it is imperative that military communities be equipped with the requisite resources to create facilitative socio-ecological conditions that foster the development and maintenance of MF resilience.

### 1.1. Rationale for Review

While international leaders agree that building resilient military members and their families is highly important for national security, the evidence on effective resilience-building programs/interventions in MFs continues to be an open scholarly question. Firstly, currently, there is no clear evidence to indicate what programs and interventions build resilience in MFs, with respect to family member age and timing on the deployment cycle. Secondly, help from service providers who are either (1) professionals—trained, licensed, registered clinicians and specialists—(2) peers—trained persons who have shared MF lived experience, but are not professionals—or (3) lay persons—trained persons who do not have MF lived experience and are not professionals—may influence MF strengthening outcomes. Specifically, we do not yet know what service provider groups—professionals, peers or lay persons—deliver programs. These programs and their deliverers may or may not be helpful to MFs. Indeed, little is yet understood about the evidence on resilience-building programs and the helpers who deliver them to MFs, considering infants to senior adults. Last, the scholarly understanding on the interaction between resilience-building and presence of adversity at various points on the deployment cycle and at the time of program delivery is virtually unknown. Thus, several questions about MF-centred resilience-building programs remain unanswered. For this reason, as a first step to contribute to the scientific body of knowledge, this review examines studies published from database inception through 2023 with a resilience-building focus specifically in MFs. The results of this review will contribute towards a greater understanding of possible programming or intervention as a gateway to MF-specific mental health support, ultimately resulting in less morbidity and mortality from moral injury or loss. To address these gaps, our review focused on identifying family-centred resilience-building programs provided to MF, including military serving members, veterans, children, youth, and civilian partners or spouses, and other family members. We describe the evidence in this research realm. At the time of this scoping review, there was no scoping review that described resilience-building programs and interventions specifically on MF members from infancy through to old age. This understanding could help service providers and decision-makers to either select or develop evidence-based programs and interventions to better serve specific age groups within MFs and guide policy regarding resilience-building programming.

### 1.2. Review Objectives

The aim of this scoping review was to explore what programs build resilience in MFs. More specifically, the primary objective was to identify programming that aims to build resilience in military members or veterans and their families. The secondary objectives were twofold: to describe the evidence that supports resilience-building programming, including what service provider groups delivered the program, and to explore whether adversity is described as a barrier or facilitator in resilience-building for MFs across the deployment cycle.

## 2. Materials and Methods

This section begins with the population of focus, followed by the inclusion and exclusion criteria for articles retained in this review.

### 2.1. Participants

The population of focus in this scoping review was military members: active-duty personnel, reservists, veterans, and their families, who included spouses or partners, parents and natural (unpaid) caregivers (military or civilian), and children and youth—single children, siblings, biological, adoptive and fostered children. Of note, service providers (SPs) were also an eligible population when they delivered a military-focused service and were participants in a program that focused on building their resilience. Therefore, this review considered studies that included a wide variety of military family members and those who offer services to them. We followed the Sager Guidelines for sex and gender descriptors to guide our reporting of sex and gender differences in this review [65].

### 2.2. Article Eligibility Criteria

Eligible studies had to meet all of the following criteria.

1.Population

We included peer-reviewed publications on studies of military families, broadly defined as including any married or cohabiting couples, spouses or partners, elders, single parents or co-parents (whether cohabiting or living separately), and children from birth to young adulthood who were dependent on their parents or live at home. Excluded were articles focused on populations other than MFs or non-peer reviewed publications, e.g., commentary or theoretical without an empirical study.

2.Intervention

We considered all programs, services and interventions identified by the search if they aimed to build resilience, through a training, webinar, or education session/series or the like. Studies reporting solely on SPs that were paid caregivers like external family help rather than natural, unpaid caregivers were excluded.

3.Outcomes

To be included, a study had to use a defined outcome relating to resilience. We accepted various terms used in the literature for resilience/resiliency, self-regulation, adjustment, overcoming adversity, adaptation, adaptability, and coping, to name some examples. The Appendix A detail the search terms used. Examples of interventions that we did not deem resilience-building in nature included articles too narrow in focus such as programs targeting military member reactions to on-the-field acute stress, individual responsivity to rapid or immediate stressors, or military task-focused outcomes. Articles written in a language other than English were also excluded.

### 2.3. Article Exclusion Criteria

To ensure consistency, articles were excluded if their descriptions did not follow our definition of resilience-building services. For example, studies where SPs mainly delivered diagnostic, assessment, evaluation, treatment of a condition without a resilience component were excluded. Specifically, articles that met any of the following criteria were excluded from the present scoping review:Non-peer-reviewed publications, e.g., commentary or theoretical without an empirical study;Articles focused on populations other than MF;Articles not studying a resilience-building program for MF, such as those too narrow in focus, e.g., programs targeting military member reactions to on-the-field acute stress, individual responsivity to rapid or immediate stressors, or military task-focused outcomes;Any studies referring to paid caregivers like external family help rather than natural, unpaid caregivers;Articles written in a language other than English.

This review was conducted according to the System for the Unified Management, Assessment and Review of Information (SUMARI) [66]. We examined published, peer-reviewed articles related to resilience-building programming provided to MF. All eligible studies that were found were analysed. Namely, we adopted a wide definition of resilience-building programs to include self-regulation, adaptability, and overcoming adversity, to name a few examples (Appendix A). Programs, interventions or services were considered when the program aim was in part or in whole to impact resilience in military family members.

### 2.4. Concepts and Context

In this section, the review context is preceded by an introduction on the two central review concepts: resilience-building and programming.

#### 2.4.1. Concept 1: Resilience-Building

The primary concept in this scoping review was resilience-building. For the purposes of the search strategy, resilience-building was defined as an interaction between a military member/service provider, veteran, family member—partner, parent, child or youth—and a service provider who could be a professional, clinician, researcher, therapist, peer, or lay person; any of whom aimed to build resilience in one or several military family member(s). This interaction could be direct with the service provider or indirect through engaging with material designed by researchers or SPs.

#### 2.4.2. Concept 2: Programming

The second concept was programming defined as a service, intervention, training, webinar, seminar, session, support group or the like. For the purposes of this review, programming referred to whether the resilience-building service was delivered through any interventional or educational means and was provided by one of three service provider groups: (1) formally trained professionals—occupational therapists, psychologists, social workers, medical doctors, or other clinicians—(2) peers—people who were not professionals but were formally or informally trained with shared lived experience such as veterans, inactive military members, and military family members—and (3) lay persons—people formally or informally trained who are not professionals and are without shared lived experience.

#### 2.4.3. Context

Our review considered resilience-building programming when provided in any setting—hospitals, clinics, or community-based settings. No geographic limitations were placed on this review, given that the intent was to explore military family-centred resilience-building programs across all settings and locations. This review included peer-reviewed studies of programs aiming to build resilience. Evidence was collected to identify the provider group and overarching program aim to summarise program impact and to determine whether adversity was interpreted as either a barrier or facilitator or both in relation to resilience-building for MF. Further, data were collected on the timing of program delivery and considerations related to the deployment cycle: prior to deployment, active-duty with or without deployment, or reintegration/transition back to family life. Next, source types provide examples of peer-reviewed articles considered for this review.

### 2.5. Type of Sources

This scoping review considered research that provided information on military family-centred resilience-building programs. We considered all articles published in English from the inception date of databases. Study designs included randomised controlled trials, non-randomized controlled trials, quasi-experimental designs, before and after studies, prospective and retrospective cohort studies, case–control studies, and analytical cross-sectional studies. Also, this review considered including descriptive observational study designs such as case series, individual case reports and descriptive cross-sectional studies. Qualitative studies were additionally considered, including, but not limited to, designs such as qualitative child/youth/family narratives or family-reported outcomes, phenomenology, grounded theory, ethnography and qualitative description. Mixed-methods studies using both qualitative and quantitative methods were also considered. In addition, systematic, scoping and literature reviews that met the inclusion criteria were considered. We excluded secondary studies and non-peer-reviewed publications such as theoretical reports, commentaries and conceptual works without an empirical study design.

A preliminary review of the literature by the lead author (M.L.H.) helped identify initial search terms (keywords) on the topic. The population, concept, and context framework [67] then helped inform the search strategy described earlier. Several authors (M.L.H. and P.R.S.) worked together to create and refine the search strategy, and then the search strategy was carried out by a health sciences library on the following databases from inception until December 2023: CINAHL Plus with Full text, Military and Government Collection, Health Source: Nursing/Academic Edition, Education Research Complete, ERIC, Open Dissertations, Criminal Justice Abstracts, Child Development & Adolescent Studies (all EBSCO databases), MEDLINE (Ovid), Embase (Ovid), APA PsycInfo (Ovid), and Social Services Abstracts (Proquest). The complete search strategy can be found in Appendix A.

### 2.6. Scoping Review Protocol Registration

Protocol data and materials were registered and made publicly available in the Open Science Framework and are accessible at https://osf.io/u2jge/ (accessed on 10 November 2022). The protocol abstract word count was 250, and the total minimum manuscript word count was 4000.

### 2.7. Article Selection, Data Extraction and Charting Process

Following the search, all citations were uploaded into EndNote version X9 (Clarivate, Philadelphia, PA, USA), and they were imported into Covidence and duplicates were removed (Covidence, Melbourne, Australia). The first two authors (M.L.H., J.M.T.) independently screened the article titles and abstracts published in English for eligibility through assessment against the inclusion criteria. Then, any disagreements that arose between the reviewers at any stage of the article selection and screening process were resolved through discussion with a third reviewer (C.A. or M.M.). Next, the full texts of potentially relevant studies that we retrieved were assessed against the eligibility criteria. Reasons for excluding full-text studies were recorded and reported in this scoping review using the Preferred Reporting Items for Systematic reviews and Meta-Analyses extension for Scoping Reviews (PRISMA-ScR) framework [68]. After article screening and inclusion were completed, the reference list of all eligible studies was also checked for additional eligible articles.

Three authors (M.L.H., J.M.T and M.M.) had developed a standardized data extraction form to extract study characteristics (Appendix A). The standardized form was pilot-tested by these three authors using the first five listed studies, in alphabetical order by lead author surname for each article. Extracted terminology and concepts were summarised using tables. Specifically, reviewers worked independently to extract study details (M.L.H., J.M.T.). A third independent reviewer reviewed data extraction and resolved any conflicts that arose (CA). The data extracted included specific details about the population, concepts, context, research methods and key findings relevant to the scoping review research questions. Limited time and resources did not allow us to contact authors of research articles and other literature sources to request additional or missing data, as proposed in the original protocol [69].

## 3. Results

This scoping review examined programs pre-deployment, post-deployment and during reintegration back into civilian life for building resilience in MF. The findings are based on 32 studies that used a wide range of research methodologies with a combined sample of 14,597 participants, ranging from 9 to 7309 participants per article. Below, we present a brief summary and discussion of the results for each outcome assessed in this review. Our reviewed studies investigated 22 programs with the aim to build resilience in MFs. We set out to meet three objectives in the present scoping review: (1) identify programming that aims to build resilience in military members or veterans and their families, (2) describe the evidence that supports resilience-building programming, including which service provider groups delivered the program, and (3) explore whether adversity is described as a barrier or facilitator in resilience-building. To answer the three review questions, we describe each study and program/intervention characteristics and details, followed by an exploration of adversity as a barrier or facilitator to resilience. This section begins with added observations about the search terms and later also reports on attrition.

### 3.1. Search Terms and Process

Search terms that were initially included such as fitness, hardiness, and grit were found in articles that concentrated solely on training active-duty or pre-deployed SMs, revealing that these terms likely are not representative or of relevant to MFs (Appendix A). This search initially identified 4050 articles that yielded 1317 deduplicated articles, for which the abstracts and titles were screened for relevance. Of these remaining articles, 299 were assessed for eligibility through full-text review, and ultimately, twenty-four studies met the search and eligibility criteria. Reference lists from these eligible articles were then examined, and we found 19 additional relevant studies for this scoping review, of which five were deduplicated and removed. Further, while four other peer-reviewed published articles were identified through expert opinion and considered for inclusion, none met the search criteria. We thus retained and analysed a total of 32 manuscripts that studied outcomes (resilience or resilience-related constructs) in military family members at various time points on the deployment cycle. One review and thirty-one other empirical studies were identified. Figure 1 presents the Preferred Reporting Items for Systematic Reviews and Meta-Analyses (PRISMA) flow diagram for our review. The diagram reflects included and excluded articles.

Appendix A present three tables of results. Appendix A summarise objectives, main characteristics and outcomes for all studies that met our search criteria and were reported in the literature since database inception, as well as general level of evidence for each study. Of note, the level of evidence assessment was not a critical appraisal; thus, the authors opted for a high-level definition from a national source. Appendix A briefly presents each program/intervention found in the reviewed studies.

### 3.2. Study Characteristics

The eligible articles revealed expected characteristics such as their geographic region, research methodologies and populations of focus, yet they also rendered unexpected characteristics, particularly on their widely varied SPs. As expected, the vast majority of the included studies (*n* = 29, 91%) were conducted in the United States, with a few located in Japan. The smallest number of studies were completed in either Canada or South Korea (Appendix A). The reviewed articles were published from 1999 to 2023, with the majority published more recently, between 2016 and 2023 (Figure 2).

Half of the articles focused on parents, parents-to-be and parents and their child or children, and the other half concentrated on other adult populations—partner dyads/spouses, active-duty military members or veterans, chaplains or SPs, or civilian caregivers of veterans (Table 1). Among these adult-focused studies, few studies included decision-makers in their sample. To our knowledge, no studies concentrated on paid caregivers like external family help rather than natural, unpaid caregivers; however, some articles, might have included both natural and paid caregivers. While investigations are emerging on teens self-reporting outcomes and on observational studies of school-age children, none centred on infants, toddlers or preschoolers alone. A minority of articles reported studies on young children and either civilian parents or teachers (*n* = 2, 6%), or on young children alone (*n* = 1, 3%). Rare were articles that concentrated on engaging the entire family—military and civilian parents and their children/youth. Two thirds were rated at a level-two evidence (63%), and the other third were either level-one (22%) or level-three (15%), respectively (Appendix A).

The majority of the studies reported the type of service provider who delivered the program (*n* = 29, 91%). Almost half of the program facilitators were clinicians or licensed professionals, and about the other half were either trained peers or lay persons, mixed SPs/facilitators, or self-directed by participants (Table 2). For example, in one study, adolescents were trained to deliver the program to their peers. In general, SPs included mental health providers, chaplains, occupational therapists, social workers, counsellors, nurses, physicians, and hospice workers. Table 2 summarises the study and program/intervention characteristics in aggregate form including geographic spread, study setting, program type and provider type. See Appendix A for figures that present additional details on study characteristics.

Virtually absent were studies that aimed to build multisystem resilience in broader societal contexts to influence decision-making or policy. Mental health, emotional wellbeing, reduction in stigma, stress/distress, anxiety or depression, and intimate relationships/family communication or interactions with children in school seemed to improve in several studies [11,71,72,73,74,78,86,87,89,90,92,93,95,100]. Effective parenting/pre-parenting was also found in some studies [70,83,85,93], including reducing infant distress in a pre-birth program [93]. One study showed promise for positive parenting, yet also showed limitations in decreasing negative parenting [84]. Children’s behaviours improved in two studies [92,97], though one had few students from MF, so outcomes may not be MF-centred [97]. A few studies centred on engagement/satisfaction with promising results [79,88,96,98]. Therefore, all these studies demonstrated overall promise in positively impacting resilience-related skills in MF members. While the majority of these outcomes are relevant and important, their nature is individual-level or family system-level outcomes rather than multisystemic, considering the broader socio-ecological context governed by health and social policy in which MFs live. Indeed, rare were studies that emphasised resilience-building through a dynamic interplay between the person, environment and relationships with others in the broader community. Less than a quarter of the studies concentrated on resilience as a construct or used resilience-specific measures. Figure 4 represents resilience, resilience-related or other outcomes measured in all studies reviewed.

The studies in the reviewed articles delivered their programs or interventions mainly during post-deployment timing with variable outcomes, and some authors discussed factors influencing attrition. The majority of researchers timed their study post-deployment or when active-duty SMs were reintegrating/transitioning back into family or civilian life. No studies timed their program solely prior to family members being deployed. When timing matched pre-deployment, the study also investigated other deployment cycle timing including post-deployment or active-duty and reintegration timing, thus transitions throughout the cycle [74,90,92,93]. Table 3 summarises program or intervention timing across the deployment cycle from pre-deployment to transitioning back to civilian life or reintegration.

Most of the programs were delivered in groups, primarily in person, and the rest were about evenly split between either online only delivery or hybrid delivery including both in-person and online modes (Figure 5).

Further, several researchers either explicitly studied or reported on adherence, attrition, or otherwise commented on this topic. Adherence was strong in articles 99 and 123 where the majority of participants, couples/partner dyads, completed the interventions at home (84% and 71%, respectively). Along similar lines, satisfaction with home practice assignments significantly predicted program completion in another study with parents [75]. Many participants were found to be lost to follow-up due to transitions, deployment status, number of relocations and separations [83,92]. Additionally, some authors found a negative relationship between the number of children living at home and adherence: the higher the number of children, the lower the adherence [83]. Lower education level and positive emotional state were also factors that helped explain attrition [85]. In another study, income levels may have a positive relationship with attrition rate in fathers. The fathers who dropped out of the intervention had higher household incomes compared to those who completed the program [93]. Last, sessions that were led by peer facilitators had higher attendance compared to sessions led by another facilitator without MF lived experience [88].

### 3.3. Exploring Adversity as Barrier or Facilitator in Resilience-Building Programming

We explored whether adversity was described as a barrier or facilitator in MF resilience-building programs and found several studies in this review that either analysed adversity as a factor or discussed adversity as part of MF life. For instance, many studies highlighted deployment or reintegration as barriers [71,75,78,80,88,90,91,93]. Some authors revealed that normalisation occurred when families shared adverse experiences related to deployment or reintegration with either other military families or within their own family unit [75,80,91,93]. Other studies factored in the frequency and type of deployment, combat or non-combat, to analyse gender differences or associations with outcomes [75,78,88,90]. Namely, significant gender differences were found in behaviour, where boys exhibited increased problem behaviour compared to girls’ behaviour post-deployment [92]. In adult gender-specific group interventions/activities and youth camps with mindfulness, a sense of community connection was shared [72,76,82]. Some authors discussed overcoming adversity and, in turn, building resilience as a result of individual-environment relationship dynamics [80,100]. Only one study applied an education program that explicitly attempted to reframe adversity as an opportunity for growth [87].

## 4. Discussion

Three reviewers participated in screening articles for relevance, eligibility and selection. The inclusion and exclusion criteria that we used to select the final set of studies were determined from expert opinion, initial search and a refined search strategy. Specifically, terms like “fitness”, “hardiness”, and “grit” that were initially included in the search were found to be more relevant to active-duty or pre-deployed SMs rather than MFs. Thus, precise and context-specific search terms in this scoping review facilitated finding MF-specific articles focused on building resilience. We intentionally focused on a broad definition of ‘family’ and considered natural, unpaid caregivers in MFs rather than paid caregivers in the reviewed articles, because of known risks for their mental health. Most reviewed articles were published between 2016 and 2019, which may reflect an increasing interest in resilience-related research among MFs in recent years. The widely diverse populations in this review spanned from infants to elders, parents of children and youth to spouses, SPs, caregivers of veterans, and mixed populations. This vast range shed light on 22 existing programs that aimed to build resilience in MF communities from infancy to 86 years of age. The majority of articles studied adult populations, a few articles studied populations aged 60 years or more, or school-aged and adolescent populations with and without parent-child engagement, and rarely did articles study newborns, infants, toddlers, preschoolers or early school-age children. This main focus on adults and older children, teens and youth points to the need for evidence-based early intervention that promotes health and wellbeing in MFs with infants or young children.

While a critical appraisal was beyond the purview of this scoping review, we noted each article’s level of evidence using a high-level assessment to evaluate the general quality of the literature on MF resilience-building. Various methodological approaches were used in the included studies, including randomised control trials, quantitative methods, mixed methodologies, and qualitative methods. Overall, fair to excellent methodologies were applied using random group assignment, single blinding in at least one study, two- to four-point repeated measures designs, and surveys, interviews and focus groups, which allowed for comparison of outcome measures over time or in-depth understanding of participant perspectives, respectively. Samples varied from small to very large, which generally matched the study objectives and design. Strong methodological approaches included, on one hand, the random assignment of participants to an intervention group, comparison group or control group, such as standard care, followed by outcome and demographic comparison, and on the other hand, qualitative methodologies that revealed relevant and important themes from participant voices of various ages. These methods helped highlight similarities and differences in participant experiences, sex and gender considerations, age-specific program relevance. For instance, mental health constructs, self-regulation and communication with other MF members generally improved.

Gender differences were also revealed in numerous articles, comparing boys and girls, mothers and fathers, men and women. Gender influenced anxiety, distress and behaviour in children. Reported gender differences in these studies may be consistent with emerging evidence indicating that social media use is associated with recent trends of declining mental health for young girls [101,102,103]. Furthermore, loneliness has risen worldwide during [104] and post-COVID-19 [105,106]. Feeling alone is a potent and well-documented risk factor for early mortality [107,108] and is associated with greater mental illness [109], emotional distress [110], and suicidality [111]. Notably, a sense of belonging was revealed as a positive outcome in a few studies that either focused on gender- or age-specific groupings—women only, clinicians only, youth only. Indeed, gender-specific groups fostered a sense of community connection or belonging.

While youth ranked mindfulness as the activity that helped them form the strongest bonds with other youth, which is an important element of resilience, mindfulness and similar clinical approaches in adults were found mainly to reduce stress that alone is not central to facilitating resilience from a socio-ecological lens. Remarkably, researchers highlight that the stress demands of civilian spouses may equal or even surpass their active-duty partners [85], because civilian spouses lack the support from being embedded in a cohesive unit, they frequently lack clear information on the risk status of their loved one, and they are unable to act instrumentally on the absent partner’s behalf. Thus, social support and belonging are central to stress reduction and resilience-building. Further, our findings emphasise the value of combining mindfulness with additional approaches and activities that explicitly build resilience in a socio-ecological way, and doing so early during pre-deployment as a prevention or gateway to later help or therapy services during post-deployment, active-duty service or reintegration in civilian life. To date, the literature presents strong evidence for using mindfulness and physical activity to increase individual psychological resilience in adults [112], including military members who are pre-deployed or deployed, and in veterans who have transitioned back to civilian life. While their systematic review determined study evidence levels with robust methodology by assessing solely randomised control trials, degree of blinding and biases in studies, these reviewed studies on military populations did not include social support by family members or the broader systems in which this population lives. Building resilience in MFs goes beyond psychological resilience, relying more so on multisystemic resilience or multi-levelled cross-sectoral social support that spans the deployment cycle.

The wide range of SPs included traditional clinicians, but also community-based peers, both with lived experience and without lived experience. Some roles of providers were well documented, and others were emerging such as in peers—trained teen ambassadors—who delivered camps to fellow adolescents. Roles in mental health providers and facilitators, such as chaplains, occupational therapists, social workers, nurses, physicians, peers and lay persons likely complement one another, and suggest that leveraging transdisciplinary care adds value to respective disciplines when targeting resilience-building. Such joint program delivery in a few studies demonstrated the feasibility of pairing or grouping distinct service provider or facilitator types: lay persons with clinicians and peers with military leaders, as two examples. One article [88] compared civilian and peer capacity to facilitate MF programming, resulting in significantly greater MF participation when led by peer facilitators. This finding points to the benefits when SPs have MF lived experience. Given the frequent relocations required of MFs, it would have been pertinent to learn about navigation service delivery. Navigators help people in their help-seeking process to facilitate finding and accessing needed services. Yet, we did not identify navigators specifically in the reviewed articles. Had the search included terms like ‘relocation’, ‘moving’ and ‘navigation’, additional studies on navigation services may have been identified.

The proportion of studies that specifically focused on resilience-related constructs such as mental health conditions was more than three quarters. This finding highlights the ongoing need for trauma-informed therapies and intervention. Over time, more scholars have embraced multisystemic perspectives of the construct that also emphasise the importance of social contexts for resilience [30,59,60,61], such as the family system [37,38]. Indeed, resilience is significantly valued in the context of MFs for each family member to overcome the tumultuousness, unpredictability and cumulative effects of military life. That being said, a growing body of literature emphasises that for resilience to be built in human beings, programs must go beyond focusing on the family system, and they must work to improve the complex inter relational dynamics in multiple systems in which human beings live: individual–family, individual–community, family–SP, SP–decision-makers, decision-makers–policymakers, and policymakers–global trends [31,59,113]. Considering this broader socio-ecological context, the literature reviewed reveals a number of shortcomings. The majority of studies concentrated on improving symptoms or mental health concerns as highlighted in the study objectives and results (Table 1 and Table 2) and the outcome measures used to demonstrate impact (Figure 4). Therefore, research to date has not yet adequately addressed resilience-building in MFs from a multi-systems transformational lens.

We also learned that several potential factors influence program adherence and completion in adults, beyond the evident impact on absenteeism related to deployment. A key finding from this scoping review is the emergence of telehealth approaches—R2MR and technology application or apps—and the feasibility and acceptability of online delivery modes, even before the COVID-19 pandemic, pointing to promise for hybrid or HyFlex formats. Multiple factors were indeed found to facilitate or impede adherence and intervention completion. Adherence was strong in studies that concentrated on partnership/spousal support where activities were undertaken in the privacy of the home. A community-based rather than clinical-based setting may have made a difference between these MFs seeking help and partaking in the program, as compared to another study on spousal education where attrition was an issue, and the program was held in a veteran/military medical setting. This environmental finding highlights what context some MFs (perhaps most) may favour when participating in self-/couple-bettering programs. Other factors such as heightened number of children in the household and father’s higher income, and lower education and good emotional state may relate to participants attritting. Further, while the interventions widely varied from camps/retreats, to massage and meditation/mindfulness, to hybrid psychoeducational modules, some similarities crossed several distinct programs. For example, as expected, transitions, deployment, frequent relocations and separations were related to loss to follow-up. Adherence or attendance was consistently low for multi-time point studies that went beyond four months. Also, aligned with historically low parental attendance and engagement rates [114], parents attritted, including both mothers or fathers based on their age and type of program activities. These noncompletion findings in parenting programs highlight the pertinence of offering hybrid or HyFlex delivery with age- and sex/gender-specific activities over two to four months maximum.

Moreover, adversity faced by MFs was explored as another factor, where only one article discussed adversity as an opportunity. Other researchers explored team resilience emerging from collective experiences of military-related adversity, highlighting their bonds as protective factors [115]. Several other authors argue that when MFs shared adverse experiences related to deployment or reintegration, normalisation occurred. When normalisation happens, the effect can be deeply positive, thereby minimising stigma and creating a sense of validation and belonging. This is consistent with findings that the quantity and depth of one’s social interactions [116] and having one’s emotions acknowledged by others—particularly negative emotions—[117] are associated with greater social connectedness and interpersonal trust, which may in turn help satisfy the fundamental human need to belong [118]. In psychotherapy, adopting an empathic, accepting, and validating stance toward client emotions is widely recognized as essential for creating a safe, helpful, and healing therapeutic relationship [119,120,121,122,123,124], with one study finding that client perceptions of therapist validation were negatively correlated with post-session negative affect [125]. Similarly, it is possible that resilience programs that create safe, normalizing environments may reduce negative affect and potentially yield additional resilience-building benefits for participants. Namely, evolution has dictated that safe and supportive relationships naturally regulate our social brains [126].

Last, most studies primarily timed their program or intervention post-deployment or during transitions back to civilian life, highlighting significant needs for support during MF separations and reunifications. Indeed, reintegration back to civilian life is among the greatest challenges for military SMs and veterans, as well as for family members. So, such therapies and interventions as described and studies in the reviewed articles will continue to be valuable and needed for MF. However, the focus on post-deployment timing also highlights a pre-deployment gap in services. For example, could a program foster help-seeking when MFs need it by helping to prevent PTS, mitigate mental health concerns and rehabilitate moral injury in children, civilian MF members, and military members and veterans, wherever they are on the deployment cycle? This remains an open question that is in need of further study.

In conclusion, to enable MF leaders, policymakers and decision-makers to select, fund and make available evidence-based resilience-building programs and interventions, it is essential to reliably collect and measure resilience in an authentic and comprehensive manner to facilitate effective strengthening of MF members at any age. The wide range of said ‘resilience’ programs and interventions, the different resilience definitions and the limited consensus in service providers and facilitators make it challenging to establish which approach is the most effective. Therefore, the existing research on resilience-building programming has resulted in a piecemeal evidence base that is challenging for deciders and service providers to differentiate.

### Implications for Clinical Practice and Future Research

Our findings have implications for practice, policy, and research in supporting the resilience of MFs. There is value in further exploring resilience-focused program effectiveness and impact when delivered by either clinicians, peers or lay persons, or a combination of these service provider or facilitator groups. Also, incorporating search terms related to ‘relocation’, ‘moving’ and ‘navigation intervention’ may render an even wider variety of SPs and programs, including navigators and navigator-delivered services in another review. Additional potential directions for future research in the field of MF resilience include better understanding sex and gender differences by systematically incorporating such analyses in reviews and studies. Lastly, we found that there exists the need for resilience-building programming that centres on resilience in MFs considering more consistent use of resilience-specific measures rather than measures of resilience-adjacent constructs, and programs that work on building collective resilience in more than MF members and SPs, including leaders, decision-makers and policymakers.

To advance resilience-building for MFs, future research should broaden its focus beyond individual and family levels to include multisystem resilience, examining how interactions with social and environmental factors impact resilience. There is a need for more studies that use resilience-specific measures to accurately assess resilience-building program/intervention effectiveness. Incorporating qualitative research will provide deeper insights into personal experiences and the development of resilience. Additionally, research would benefit from involving decision-makers and paid caregivers to gain a comprehensive understanding of support systems. Addressing factors contributing to attrition and low adherence, such as family size and household income, is crucial. Studies would benefit from exploring the impact of resilience programs at different points in the deployment cycle, including pre-deployment, and evaluate various delivery modalities—whether in-person, online, or hybrid. Reframing adversity as an opportunity for growth and expanding research to diverse geographic and cultural contexts would also enhance the development of effective, comprehensive, and culturally sensitive resilience-building programs for MFs.

## 5. Conclusions

Although we need to exercise caution in interpreting these findings because of the high heterogeneity in studies, these findings nonetheless appear to help inform the provision of military family-centred resilience-building programming. The main takeaways from this scoping review underscore the importance of continued research in this area to better understand sex/gender differences in MFs and improve MF resilience-building, including early intervention with young children from birth to preschool ages. Program delivery both in-person and online may be preferable, particularly to allow for MFs to participate from the privacy of home, anywhere in the world. Other facilitators of preventing dropouts include trauma-informed programming, flexible/asynchronous learning opportunities, gender-specific and youth-centred groupings. This review also emphasises the value of transdisciplinary service provision when focused on MF-centred resilience-building. More research and program development are needed to effectively address these factors either by enhancing existing programs or designing new ones to build resilience in MFs at any time throughout the deployment and reintegration cycle. Last, recognizing adversity as an opportunity for normalisation may help promote MF resilience, wellbeing and belonging.

### Limitations

In this review, the coding and management of data in articles using qualitative methodologies were not analysed, nor was conducting full critical appraisals of articles using quantitative methodologies or statistical analyses. This narrower scope allowed for the review to be completed in a shortened time frame with broad article inclusion criteria. Such a broad focus aligned with the scoping, descriptive nature of this review. Also, articles that did not meet eligibility criteria or did not use our search terms may not have been found yet may have been valuable in building resilience. Given that the majority of studies have been conducted in the United States, where military legislation and policy differ significantly from other countries, the US-based findings may limit their generalisability to other countries. Further MF resilience-building research rather than on resilience-related constructs, particularly in nations other than the US, are needed.

## Figures and Tables

**Figure 1 ijerph-21-01378-f001:**
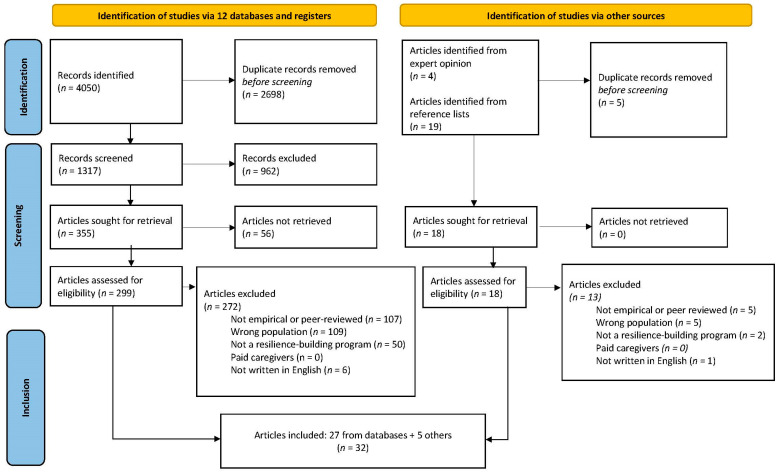
PRISMA diagram of included and excluded articles in this review.

**Figure 2 ijerph-21-01378-f002:**
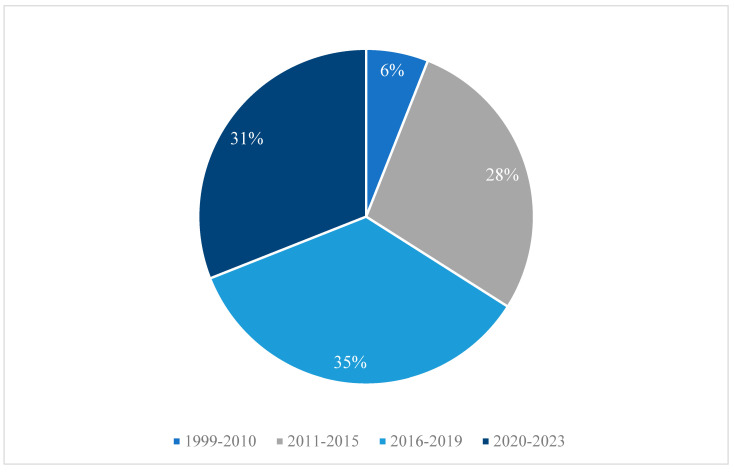
Year of publication for articles reviewed. Note: 1990–2010: articles: [70,71]; 2011–2015: articles: [72,73,74,75,76,77,78,79,80]; 2016–2019: articles: [11,81,82,83,84,85,86,87,88,89,90]; 2020–2023: articles: [91,92,93,94,95,96,97,98,99,100]. Just over a quarter of the studies used a randomised control trial or randomisation approach (*n* = 9, 28%). The majority used other quantitative or mixed methodologies, and few used qualitative methodologies alone (Figure 3).

**Figure 3 ijerph-21-01378-f003:**
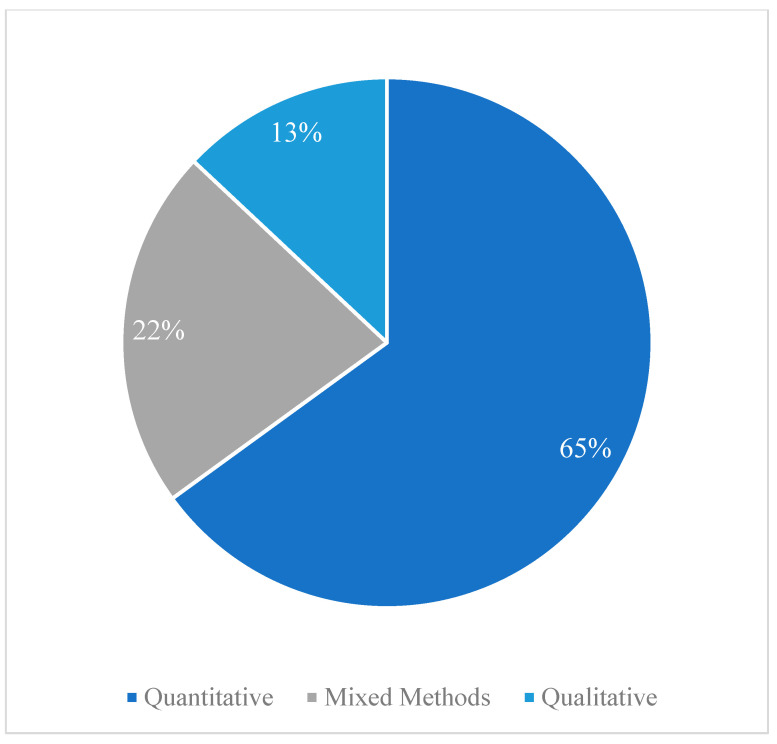
Methodologies used in reviewed studies. Note: Quantitative: articles [11,70,71,74,75,77,78,83,84,85,86,87,88,89,90,91,92,93,94,97,100]; Mixed Methods: articles [72,73,76,79,81,82,99]; Qualitative: articles [80,95,96,98].

**Figure 4 ijerph-21-01378-f004:**
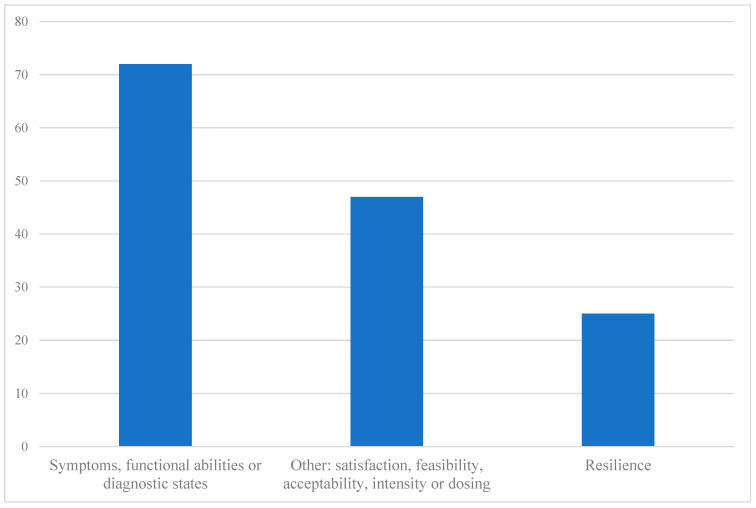
Outcomes measured in reviewed studies. Note: Symptoms, functional abilities or diagnostic states [11,71,72,73,77,78,80,81,82,83,84,85,86,87,89,90,91,93,94,97,98,99,100]; Other: satisfaction, feasibility, acceptability, intensity, or dosing [73,74,75,76,77,79,80,82,88,91,92,95,96,98,99]; Resilience [71,90,91,92,94,95,96,97].

**Figure 5 ijerph-21-01378-f005:**
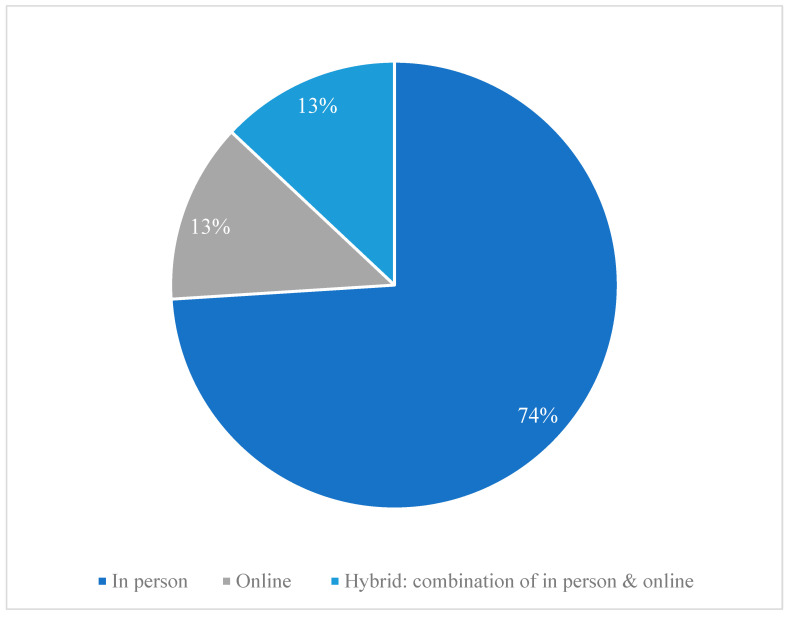
Program or intervention delivery mode in reviewed studies. Note: In person: [11,72,82,91,92], [74,75,83,84] ^a^, [70,71,76,77,78,79,80,85,87,89,90,94,95,96,97]; Online: [81,86,93], [98] ^b^; Hybrid: [73,88,99,100]; ^a^: experimental group was in person and control group was online; ^b^: delivered on Facebook.

**Table 1 ijerph-21-01378-t001:** Main family member populations of focus.

Population	*n* (%)	Identifier
Parents: 16 (50)		
parents only	4 (13)	[75,84,85,88]
expecting mothers	2 (6)	[70,90]
expecting parent dyads expecting and infants	1 (3)	[93]
fathers	1 (3)	[82]
parents/teachers and children age 3–5 years	2 (6)	[77,97]
parents and children age 6–12 years	5 (16)	[11,78,83], [89] *, [80]
parents and teens age 13–17 years	1 (3)	[91]
Other Family Members: 16 (50)		
spouses or partners	4 (13)	[71,73,79,86]
service providers	1 (3)	[74]
caregivers of Veterans	1 (3)	[81]
infants	0 (0)	
toddlers, preschoolers	0 (0)	
children	1 (3)	[92]
teens	3 (9)	[76,94,100]
mix of two or more adult populations	6 (19)	[72,87,95,96,98,99]
	32 (100)	

*: Children’s ages are unreported, though the authors refer to children in school ages.

**Table 2 ijerph-21-01378-t002:** Descriptive summary of study and programs/interventions.

**Study Characteristics**	***n* (%)**	**Identifier**
Geographic Spread		
Regional: one or more sites within a single state or province	16 (50)	[71,72,79,81,82,83,84,86,88,89,90,91,94,98,99,100]
National: multiple sites in two or more states or provinces	12 (38)	[73,74,75,76,78,80,85,87,92,93,96,97]
International: multiple sites in two or more countries	4 (12)	[11,70,77,95]
	32 (100)	
Study Setting		
Military Base or Reserve/ Military Installations	13 (41)	[11,70,71,75,77,78,80,83,87,89,92,95,96]
Home-based	7 (22)	[73,81,82,86,88,93,99]
Community	4 (13)	[72,84,85,100]
Preschool / Schools	3 (9)	[74,94,97]
Medical Settings	2 (6)	[79,90]
Outdoors/ Camps	2 (6)	[76,91]
Post-secondary Institutions	1 (3)	[98]
	32 (100)	
**Program Characteristics**	***n* (%)**	**Identifier**
Program or Intervention Type		
Service provision or therapy and education	25 (78)	[11,70,72,75,76,77,78,79,80,81,82,83,84,85,87,88,89,91,92,93,94,96,97,99,100]
Education	5 (16)	[71,73,86,90,98]
Service provider training	2 (6)	[74,95]
	32 (100)	
Provider or Facilitator Type		
Clinician, professional, intern/student being trained in a profession	15 (47)	[11,71,73,74,76,77,78,79,81,89,94,95,96,97,98]
Mixed	5 (16)	[72,83], [87] ^a^, [70,80]
Peer as military family member with lived experience	4 (13)	[75,88,90], [100] ^b^
Lay person, not a military family member	3 (9)	[92] ^c^, [82,99]
None, self-directed by participant(s)	2 (6)	[86,93]
Unreported	3 (9)	[84,85,91]
	32 (100)	
All percentages are rounded		

^a^: trained staff at REBOOT headquarters, thus may be Chaplains; ^b^: often a high-ranking military officer; ^c^: with access to a clinician or professional.

**Table 3 ijerph-21-01378-t003:** Deployment cycle timing in reviewed programs.

Population	*n* (%)	Identifier
Pre-deployment *	0 (0)	
Post-deployment or active-duty	10 (31)	[11,70,71,77,78,80,82,83,89,94]
Reintegration	6 (19)	[73,75,79,86,91,99]
Mixed **	12 (38)	[72], [92] *, [93] *, [74] *, [84,85,87,88,95,96], [90] *, [100]
Unreported	4 (12)	[76,81,97,98]
	32 (100)	

*: While no studies centred on pre-deployment only, four timed their program or intervention during pre-deployment or other deployment cycle timing. **: Mix of two or more deployment cycle timing.

## Data Availability

The original contributions presented in the study are included in the article/Appendix A, further inquiries can be directed to the corresponding author.

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
