# Peer review of "Military Family-Centred Resilience-Building Programming Across the Deployment Cycle: A Scoping Review"

_ijerph, 2024, doi:10.3390/ijerph21101378_

Round 1

Reviewer 1 Report

Comments and Suggestions for Authors

This is a timely and appropriate paper. The authors cope valiantly with the various definitions of terms required for the review, such as 'resilience' and 'military family', made more complicated by the nuances used across the nations likely to contribute to the literature.

The findings are important although almost exclusively relate to the USA. This could be strongly emphasised in the abstract.

Despite the efforts to provide definitions, an early requirement needs to be stated in the title, abstract and introduction if this study reflects regular Service personnel or regular and reserve, once again recognising the various definitions across the countries involved. The term 'active-duty' personnel is not defined and in some countries may or may not include full time or high readiness reserves. This would include reservists' MFs  involved in resilience building programmes.

Reservists and reservists' families are potentially more vulnerable to the stresses of service, deployment and of course transition between service and civilian life multiple times without becoming veterans. If the review has been able to shed light on this and if resilience programmes have been designed with this in mind it would enhance the paper.

Otherwise the paper is well written with sound outcome, discussion and limitations all included.

Author Response

Reviewer 1:

Comments and Suggestions for Authors

This is a timely and appropriate paper. The authors cope valiantly with the various definitions of terms required for the review, such as 'resilience' and 'military family', made more complicated by the nuances used across the nations likely to contribute to the literature.

The findings are important although almost exclusively relate to the USA. This could be strongly emphasised in the abstract.

Our Response:

In line 21 of the abstract in the ‘Findings’ section we have revised our statement of USA publications to read: “The vast majority of included studies  (91%) were conducted in the United States.”

Despite the efforts to provide definitions, an early requirement needs to be stated in the title, abstract and introduction if this study reflects regular Service personnel or regular and reserve, once again recognising the various definitions across the countries involved. The term 'active-duty' personnel is not defined and in some countries may or may not include full time or high readiness reserves. This would include reservists' MFs  involved in resilience building programmes.

Reservists and reservists' families are potentially more vulnerable to the stresses of service, deployment and of course transition between service and civilian life multiple times without becoming veterans. If the review has been able to shed light on this and if resilience programmes have been designed with this in mind it would enhance the paper.

 The reviewer makes an excellent point regarding the distinction between active service and reservists. In our scoping review, our intent was to cast the widest possible net to include all military members - including reservists. Thus our results are inclusive of reservists and their families. Unfortunately our reporting of results is limited by the ways in which members were described in the source material. To highlight the inclusive nature of our review we have edited the abstract to state that we view military families as including “active service members, reservists, veterans, their families” (lines 10-11). We have also added a clarifying statement in the introduction to note that our definition of military families “is inclusive of active service members and reservists“  (line 37-38)

Otherwise the paper is well written with sound outcome, discussion and limitations all included.

Reviewer 2 Report

Comments and Suggestions for Authors

This is an interesting manuscript that addresses an important issue on the field of military family medicine: what is the state of resilience enhancing programs for military families.  Several considerations limit enthusiasm for the manuscript in its current form.

First is the consideration of the construct of resilience itself.  There is no inclusion of how resilience is defined and measured.  This is a critical omission.  The authors need to indicate a consensus or lack thereof in theoretical conceptualizations and measurement of the construct prior to considerations of programs.  

As a related note, the authors misspeak in lines 11-12 as well as 148-149.  These examples are not in fact a definition of resilience, which is absent from the manuscript.  Rather, the authors implicate that the construct of resilience can be understood from multiple perspectives, ranging from the individual to complex systemic factors. As the authors indicate that a paucity of such multi systemic resilience building programs exist in military family work, it’s hard to justify the allusions to their upcoming efforts.  Perhaps published examples of such programs in other communities would increase the justification. 

Finally, the authors fail to include a developmental perspective.  Critical in engaging children from any population is the consideration of developmental stages.  It would be valuable if the authors could conceptualices what would be considered resilient behavior across developmental stages, incorporating the clear gender differences observed in their present review.

Author Response

Reviewer 2:

Comments and Suggestions for Authors

This is an interesting manuscript that addresses an important issue on the field of military family medicine: what is the state of resilience enhancing programs for military families.  Several considerations limit enthusiasm for the manuscript in its current form.

First is the consideration of the construct of resilience itself.  There is no inclusion of how resilience is defined and measured.  This is a critical omission.  The authors need to indicate a consensus or lack thereof in theoretical conceptualizations and measurement of the construct prior to considerations of programs. 

Our Response:

This is an important point. We have added a definition (beginning line 184): “:Therefore in this study, we define resilience as a system’s process of adaptation (e.g., individuals, groups, organizations) following adversity or risk exposure. Resilience is a dynamic process influenced by socio-ecological system interactions and available resources, which function as protective factors and lead to improved outcomes when coping with and overcoming adversity [60]”.

As a related note, the authors misspeak in lines 11-12 as well as 148-149.  These examples are not in fact a definition of resilience, which is absent from the manuscript.  Rather, the authors implicate that the construct of resilience can be understood from multiple perspectives, ranging from the individual to complex systemic factors.

As the authors indicate that a paucity of such multi systemic resilience building programs exist in military family work, it’s hard to justify the allusions to their upcoming efforts.  Perhaps published examples of such programs in other communities would increase the justification.

We agree that allusions to our upcoming efforts are premature and should not be included in this review. Consequently, we have removed such allusions in the manuscript.

Finally, the authors fail to include a developmental perspective.  Critical in engaging children from any population is the consideration of developmental stages.  It would be valuable if the authors could conceptualices what would be considered resilient behavior across developmental stages, incorporating the clear gender differences observed in their present review.

We appreciate the reviewer’s comment and agree that developmentally appropriate interventions are critical. To be reflective of the most current advances in resilience science we have centered a social ecological approach. As we note ( beginning on line 182),”Overall, multisystemic viewpoints of risk for developing psychopathology and resilience have become prominent in contemporary developmental psychology and resilience science [40, 58, 61–62].” and further discuss (beginning on line 742): “To advance resilience-building for MFs, future research should broaden its focus beyond individual and family levels to include multisystem resilience, examining how interactions with social and environmental factors impact resilience”.

Since the field is moving on from an individual view of resilience and the so called ‘invincible kids’ that permeated earlier studies, we are highly hesitant to place a focus on ‘resilient behaviours’ as doing so implicitly situates resilience within children while deemphasizing the critical importance of contextual variables. 

Reviewer 3 Report

Comments and Suggestions for Authors

1.  The paper is very interesting and sound.  I only offer a few articles/papers that might add to it, maybe they were considered but not included for sound reasons.

2.  However, these papers/articles dealt with programming designed to help USA military families be resilient.

Aronson, K.R. et al.  (2018).  Military family advocacy in the US Army: Program service outcomes and family participation.  Journal of Child and Family Studies, 27, 218-226.

Taft, C. T., et al.  (2016).  Strength at Home Couples program to prevent military partner violence: A randomized controlled trial.  Journal of Consulting and Clinical Psychology, 84(11), 935-

Sandburg, W. E., et al.  (1988).  Enrichment and counseling activities in US Army Family Life Centers.  Military Chaplains' Review, 16, 63-67.

The above reference is included because we found higher divorce rates after overseas deployments in more than one study but the Army told us not to publish this negative sort of result.  We did many years later, though.  The point is that the military can censor a lot of "bad news" research whether on family needs or program outcomes, biasing results of ANY study in a favorable (for the Army brass) direction.  The paper might take care to mention the possibility of such bias.  The Pentagon paid the RAND corporation about a million dollars to refute our research and they found what the Pentagon wanted by restricting their analysis of post deployment divorce rates to the day the soldiers returned from overseas, BEFORE they had a chance to complete any upcoming divorce.  RAND concluded that overseas combat deployments were not even "stressful" for soldiers and their families because divorce rates were no greater.  Later, the researchers had to admit that such deployments were indeed "stressful."

Schumm, W. R., et al.  (2019).  Military family research: Methodological lessons learned, often the hard way, updated with a eulogy for Dr. Bruce Bell.  Archives of Psychology, 3(3), 1-22.

Doing research on the military and/or military programs is not easy and fraught with research landmines.  This paper could be mentioned as a caveat to the risks of error/bias in reported results due to the serious methodological problems.

Feel free to ignore any of these matters if you wish.

Author Response

Dear Reviewer,

Thank you for your comments. I have attached the coverletter.

Best Regards.
